The Author(s) *BMC Pregnancy and Childbirth* 2017, **17**(Suppl 2):350

**RESEARCH**     

# Abortion as empowerment: reproductive rights activism in a legally restricted context

Julia McReynolds-Pérez

## Abstract

**Background:** This paper analyzes the strategies used by activist health professionals in Argentina who justify providing abortion despite legal restrictions on the procedure. These "insider activists" make a case for abortion rights by linking pregnancy termination to a woman's ability to exert agency at a key point in her reproductive life, and argue that refusing women access to the procedure constitutes a grievous health risk. This argument frames pregnancy termination as an issue of empowerment and also as a medical necessity.

**Methods:** This article is based on ethnographic research conducted in Argentina in 2013 and 2015, which includes in-depth interviews with abortion activists and health professionals and ethnographic observation at activist events and in clinics.

**Results:** During the period of my field research, the medical staff in one clinic shifted from abortion counseling, based on a harm reduction model, to legal pregnancy termination, a new mode of abortion provision where they directly provided abortions based on the legal health exception. These insider activists formalized the latter approach by creating a diagnostic instrument that frames women's "bio-psycho-social" reasons for wishing to terminate a pregnancy as medically justified.

**Conclusions:** The clinical practice analyzed in this article raises important questions about the potential for health professionals to take on an activist role by making safe abortion accessible, even in a context where the procedure is highly restricted.

**Keywords:** Abortion, Latin America, Health professionals, Diagnosis, Reproductive rights

## Background

In Argentina, as in most of Latin America, abortion is illegal in all cases except for a few narrow exceptions. Or, to be more precise, most people — doctors, lawyers, judges, and the general public — interpret Argentina's abortion law as only allowing the practice when a narrow set of extenuating circumstances are present. This article analyzes the work of activist health professionals who dramatically reinterpreted the nation's abortion law and have created a clinical practice and diagnostic instruments that legitimized and implemented their interpretation. These activists argued that access to abortion is central to a woman's ability to define and carry out her

own life plan, and that infringing upon this right has a foreseeable negative impact on a woman's health that health professionals must seek to mitigate. They redefined the legal exceptions to abortion in such a way that they cover essentially all women who seek to terminate pregnancies. My data focus on the changing practice in one government-run primary care clinic where health professionals were providing abortion services openly and free of charge. The practice innovations from that clinic were also being used in other government clinics and some activist-run clinics. Because the health professionals I focus on here were playing an activist role from within the government infrastructure, I refer to them as insider activists.

My research question is: How were these insider activists shifting the institutional response to women seeking abortion at the primary care clinic where they operate? I

Correspondence: mcreynoldsperezja@cofc.edu
Department of Sociology and Anthropology, College of Charleston, Charleston, SC 29424, USA

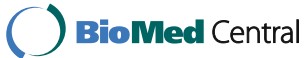

find that these insider activists created a diagnostic tool that codifies a broad interpretation of the legal health exception to justify most or all abortions as legal under existing law, in a context where this is not the typical interpretation of that law. In a reversal of the typical process of medical professionals fitting patients' medical histories to pre-existing diagnostic categories, this "legal pregnancy termination" diagnostic tool was created from diagnostic categories derived from women's narratives. Both the expansive definition of legal abortion and the fact that insider activists conform the clinic practice to the needs of their patients are ways in which these insider activists framed reproductive health provision as an issue of empowerment of their mostly poor and marginalized patients. This strategy is influencing the legal debates about abortion in ways that could have a broader impact.

### Abortion law and practice in Argentina

Even as abortion law has remained unchanged in recent decades, feminist activism and new pharmaceutical technology have created opportunities for insider activists to expand access. Some of the material presented here as background is based on my research in Argentina, which is discussed in detail below in the Methods section. This broader legal and activist context constitutes the background for the particular analysis presented in this article, of changes in practice in a government primary care clinic.

One of the openings that allowed insider activists to expand the definition of legally sanctioned abortion was a legal debate about the circumstances under which abortion is allowed. Argentina's abortion law dates to the 1920s. The law states that abortion is illegal, punishable by a range of sentences from 3 to 15 years in prison or more. The legal exceptions under which abortion is allowed include "avoiding a risk to the life or health of the mother and if this risk cannot be avoided by other means" and if the pregnancy "is the result of a rape" [1]. During the twentieth century and into the twenty-first century these legal exceptions were largely meaningless, as doctors typically refused to perform abortions without a judicial order, which could nearly never be acquired in time for the procedure to be completed before the pregnancy had run its course. In 2012 the Argentine Supreme Court ruled that all abortions that fit these legal exceptions had to be handled speedily and satisfactorily by doctors, not sent to the courts. The ruling exhorted all public hospitals to put in place procedures for determining which abortions were legal [2].

In addition to this judicial mandate to increase certain kinds of abortion access, the country has also seen the development over the last several decades of a powerful feminist movement that has made abortion legalization one of its key demands [3]. This movement gained strength since democratization in the 1980s following years of military rule and political instability. By the mid-2000s the abortion rights movement had coalesced into a National Campaign for the Right to Legal, Safe, and Free Abortion, with dozens of individuals and groups as signatories [4].

Another important opening since the 1990s was the availability of misoprostol in pharmacies throughout Latin America as a treatment for gastric ulcers. Because this pharmaceutical causes uterine contractions, women use the drug to induce abortion [5]. As is documented elsewhere [6, 7], starting in the 2000s both feminist activists and health professionals began to offer women information about inducing abortion safely at home using misoprostol. In some government health clinics, including the ones I focus on in this article, this insider activism took the form of what is called in Argentina "pre and post abortion counseling,"[1] which I will refer to simply as abortion counseling. In this practice, teams of health professionals meet with women in "before and after" appointments to advise and follow up on abortions that take place outside the clinic, with the use of misoprostol that the women procure on their own. These health care teams typically consist of some combination of social workers, psychologists, and physicians, who work together to evaluate and counsel each patient. This clinical practice was meant to ensure that if women were going to abort with misoprostol, they knew how to do it safely, yet health professionals could sidestep any possible accusations that they were providing illegal abortion services.[2]

At the clinic I focus on here, there was a shift in the way the health professionals responded to women seeking abortion between 2013 and 2015. My analysis is based on observations and interviews with insider activists at one government primary care clinic, though as I will describe later, the practice was also in use at other public and private/activist clinics. These health professionals shifted from abortion counseling to redefining women's abortions as legal pregnancy terminations based on the legal exceptions. These insider activists documented each case with a detailed justification for its legality and directly provided either misoprostol for home induction of abortion or, in some cases, performed aspiration abortions within the primary care clinic. I call this practice "legal pregnancy termination," though I recognize that what is at stake is precisely the question of whether or not this practice is truly legal under the current law.

In order to justify providing a procedure that is typically understood to be illegal, these health professionals used a diagnostic instrument to frame a woman's right to terminate a pregnancy by documenting what they describe as the patient's "bio-psycho-social" risk factors.

These health professionals consequently argued that forcing a woman to continue an undesired pregnancy to term had the potential to severely disrupt elements of her bio-psycho-social health and so denying the pregnancy termination inherently constituted a health risk. Effectively, these activist medical professionals argued that essentially all abortions are legal under the current law as they are medically necessary to protect women's health.

### Empowerment, self-determination, and reproductive rights

There is an important debate about the meaning of women's empowerment as an area that seeks to connect feminist priorities with the funding and institutional support of international development initiatives [8]. International donors and transnational governing bodies require measures of empowerment that can be quantified, compared across wide contexts, and used to determine the impact of different policies. Yet this is complicated, because women's lives are embedded in patriarchal institutions that may make women's individual choices difficult to assess. Reproductive choices are central to women's empowerment, because they are first-order, "strategic life choices which are critical for people to live the lives they want" [8]. Uberoi and de Bruyn have argued for reproductive rights as an issue central to women's self-determination [9]. They cite Anand, who writes as the UN Special Rapporteur on the right to health, arguing that laws that restrict access to reproductive health services interfere with "human dignity," which "requires that individuals are free to make personal decisions without interference from the State, especially in an area as important and intimate as sexual and reproductive health" [10].

The World Health Organization's (WHO's) definition of complete health was referenced repeatedly in government documents and by insider activists in Argentina as a basis for expanding abortion access. The WHO's Constitution lists under guiding principles: "Health is a state of complete physical, mental and social well-being and not merely the absence of disease or infirmity" [11]. Based on this definition of health, which explicitly includes social well-being, the health exception to abortion can likewise be interpreted very broadly. As reproductive rights activists in Argentina and elsewhere argue, if we take seriously the social well-being component of the definition of complete health, it becomes necessary that *all* reproductive health services be available to women, including abortion. Foreclosing women's ability to terminate a pregnancy when that is what they desire inevitably has an effect on physical, mental, and social well-being.

By most measures Argentina is a country where women's empowerment is relatively high. The country graduates more women than men from secondary level education, and it has relatively high rates of both female labor force participation and women in leadership roles in politics. This is reflected in the country's relatively high Gender Development Index of .982, which is calculated comparing women's and men's average life expectancy, schooling, and income for any given country [12]. Yet the country's Gender Inequality Index is dragged down by its high rate of maternal mortality and high adolescent birth rate relative to similar countries [13]. Therefore, Argentina is a country that exemplifies how different elements of gender empowerment do not necessarily vary together, especially depending on whether reproductive health enters into the equation. Indeed, the disparity between these two measures of gender equality likely reflects a class divide between educated, middle-class women whose options for participating in economic and political life are broad and whose education levels are as high as or higher than those of men, and poor women whose options are much more restricted.

In countries like Argentina where abortion is illegal, highly restricted, or otherwise made inaccessible, the institutional response to undesired pregnancy is disempowering, providing women with no options other than to continue the pregnancy to term. This institutional response denies women agency at a critical juncture in their reproductive lives and closes off the possibility of choosing an alternative future for their families and themselves. In Argentina, as elsewhere, these restrictive laws disproportionately affect poor women who cannot afford safe clandestine abortions in private clinics. Additionally, women's lives are more likely to be disrupted by unplanned and undesired pregnancies if their income and living situations are already precarious. As Upadhyay et al. [14] have discussed, past research has considered the effect that women's empowerment has on their fertility, but it has failed to consider that "fertility outcomes may *cause* changes in women's empowerment" [emphasis in the original]. Indeed, the inability to prevent pregnancy or access safe abortion may reflect a woman's low level of empowerment, but resulting undesired and/or successive pregnancies also have a high potential to further restrict a woman's autonomy and self-determination.

Abortion is typically a rare event in any individual woman's life. Yet *lack* of access to abortion can have dire consequences that affect the entire trajectory of a woman's life. These dire consequences in extreme cases can include mortality and morbidity from dangerous back-alley procedures. Since abortion is safer than labor and delivery, especially for women who lack adequate prenatal care, mortality and morbidity from childbirth complications could also be considered otherwise preventable health outcomes that women who are not given

access to abortion are subsequently more likely to face. But forced maternity is a much more common result that also may radically change the course of a woman's life. If undesired childbearing exposes a woman to job loss, family rejection, economic hardship, or foreclosed educational opportunities, we know based on fundamental cause theory that these social factors will ultimately have negative health effects [15]. Indeed, the analyses from the groundbreaking Turnaway Study support the argument that lack of access to abortion is demonstrably detrimental to the lives, life plans, and health of women [16–19].

### From abortion counseling to legal pregnancy termination

The clinical practice of abortion counseling was imported to Argentina from Uruguay [20]. Its proponents describe abortion counseling as a "harm reduction" strategy, borrowing language from needle exchange programs and other public health responses to illicit drug use. The idea is that criminalizing abortion does not reduce its incidence, so a public health response must seek to reduce the damages done by people engaged in that practice while withholding moral judgment [21]. Just as health professionals distribute clean needles so that intravenous drug users do not contract bloodborne diseases, in abortion counseling health professionals provide women with complete information for inducing abortion safely using misoprostol, so that if women choose to abort, they are able to do so safely.

There is a small but significant minority of health professionals in and around Buenos Aires who have been providing abortion counseling since the mid-2000s, either openly with the support or indifference of their superiors, or secretly [20]. In August of 2015 the province of Buenos Aires passed a law instituting abortion counseling at all of the province's primary care clinics for women seeking to terminate pregnancies [22]. Though the practice was already in use at some clinics and hospitals, this law provides greater legal protection for insider activists who wish to continue or expand this practice in the province.

In a small number of clinics the practice of abortion counseling had been described to me in detail when I conducted interviews in 2013. By 2015, the health professionals in one of these clinics had shifted to the practice of legal pregnancy termination. This change resulted in part from medical professionals' recognition of important limitations inherent in the abortion-counseling model. With funding from the National Health Ministry, the health professionals in one clinic completed a research report reviewing data from their abortion-counseling service [23]. The report was largely positive, describing how the service allowed women to make a truly autonomous decision about their reproductive future, free from fear of dangerous back-alley procedures. Yet the report recognized an important limitation: a plurality of patients (32%) never returned for their post-abortion appointment. Within this group there was an over-representation of women with the lowest education levels, likely suggesting that these women were poorer and more vulnerable than their counterparts who did return for the follow-up appointments. The medical staff were disturbed by the possibility that these women were unable to acquire misoprostol and so were forced to continue the pregnancy. Additionally, misoprostol taken alone has a failure rate of more than 10% [5].[3] Though misoprostol solved the problem of unwanted pregnancy safely for many women who made their way through the abortion-counseling service, for at least 1 in 10 the pills taken at the proper dosage failed to induce an abortion. Women then faced concerns about a pregnancy that was now more likely to be entering the second trimester, making other clandestine abortion options even more dangerous, and with an increased risk of birth defects should they take the pregnancy to term [24]. Perhaps the service was giving a significant proportion of women hope and then failing them.

### Institutional openings for expanding abortion access

It was in this context of frustration with the limitations of merely providing abortion counseling that insider activists at the clinic began to offer abortion services directly, eventually claiming that essentially all abortions are legal health exception abortions. It is important to understand this decision as existing in a context of legal constraints, given that the penal code is typically understood to define abortion as illegal in most cases. But there was also a level of institutional opening to the possibility of expanding the definition of legal abortion, with both national and transnational actors playing a role. In 2014 the municipality that oversees this clinic passed an ordinance giving official support to the abortion-counseling model. The practice in the clinic predated the municipal ordinance, but the local law further legitimized this practice and likely offered some protection from prosecution.

Moving from the local to the national level there is less direct, but still significant support from the National Health Ministry. When I interviewed health professionals in this clinic in 2015, the Argentine National Health Ministry had just days earlier released a *Protocol for the Comprehensive Treatment of Persons with a Right to a Legal Pregnancy Termination* [25]. In comparison to an earlier version of this document, from 2010, the definition of health exceptions presented in the new 2015 document is both broad and vague enough that it could readily be interpreted by

sympathetic parties to allow, or even mandate, the legal pregnancy terminations taking place in the clinic I describe. Though Catholic authorities expressed concern about the content of this new guide [26], health ministry officials insisted that the intent was not to change policy, but merely to bring clinical practice in line with the 2012 Supreme Court decision [27].

The new protocol begins by asserting that women's autonomy is of the utmost importance. It argues that the concept of "danger" to health "does not require the appearance of harm, but rather its possible occurrence"[4] [25]. The document also specifically warns the health professionals that are its intended audience that, "Incomplete or inadequate information, or underestimating the risk [faced by a woman seeking a legal pregnancy interruption] can result in the legal responsibility of the health professional involved" [25]. So the emphasis is on potential legal sanctions professionals face if they deny an abortion that should be allowed, rather than any legal risk to performing an abortion that might not adequately fit the legal exception. The list of authorities involved in the preparation of this new protocol includes activists and doctors engaged in efforts to legalize abortion and local reproductive rights groups funded by the International Planned Parenthood Federation [28]. The clinical practice presented here is shaped, and protected, by a wide variety of reproductive rights actors nationally and internationally, who are actively promoting the health exception as a way to expand abortion access [29].

In contrast to the practice at the handful of clinics I describe here, at most clinics and hospitals across Argentina abortion was still seen as fundamentally illegal and even cases that fit the legal exceptions were often brushed aside. Though prosecution was rare, it was not inexistent. The fear of prosecution, both for women seeking abortion and for doctors who were not supported by their superiors, was very real. Indeed, though the health professionals I describe consider abortion legal based on health exceptions in most cases, in other clinics patients were likely to face condemnation and refusal of any kind of evaluation of the legality of an abortion, no matter the circumstances. For example, at an activist clinic a doctor described the case of a 13-year-old girl who had become pregnant as a result of sexual abuse. Yet when her mother took her to a local hospital for what should have been a clear-cut legal pregnancy termination, the doctors sent the girl home with a prescription for prenatal vitamins, refusing to discuss the matter further. It was within this sometimes-confusing context of legality that the clinic at the center of this analysis had moved from abortion counseling to directly providing legal pregnancy terminations.

## Methods

This paper is based on ethnographic research conducted in Argentina primarily in 2013 and 2015. My larger research project analyzed the ways in which the availability of misoprostol was changing activist praxis and political discourse around abortion. The research I present here is based on a portion of that broader study, focusing on the role that health professionals are playing when they take on an activist role within the government health system. The larger project includes 21 in-depth interviews for which audio recordings were created and transcribed in Spanish. Each of these interviews lasted between 45 minutes to just over 2 hours. Transcripts were coded by hand. An additional 11 interviews were conducted informally and not recorded, but I took brief notes during these interviews and took detailed ethnographic notes as soon after the interviews as possible, usually the same day.

Counting both recorded and unrecorded interviews, my interviewees were 8 activists involved in pressuring the government for the legalization of abortion in some form, 9 activists involved in direct-action strategies to make abortion safer by providing women with information or services outside the formal health system, 12 health professionals who were involved in some form of abortion activism or provision, and 1 government official at a municipal health ministry. The health professionals I spoke with included one social worker, three psychologists, one nurse, and eight medical doctors (in this last category four were obstetrician/gynecologists). These professionals worked in one government primary care clinic, two government hospitals, and five activist clinics outside the government health system. Interviews were conducted in the public primary care clinic, the two hospitals, and three of the private or activist clinics. Health professionals from the remaining two activist clinics were interviewed at other locations, either at their convenience or because they were protective of the location of the clinic where they worked. Most of these were located in the metropolitan region known as Greater Buenos Aires, but I also spoke to activists who provided abortion services at clinics in the cities of Córdoba and Mendoza. I did not interact directly with patients seeking care in these clinical settings, because the focus of my research was on activism, professional authority, and political discourse. My analysis is also based on analysis of the intake documents used by clinic staff in their interactions with patients (see Additional files 1 and 2).

The research I present here focuses on the changes that I observed in one public primary care clinic in the province of Buenos Aires, where I conducted formal interviews with a social worker who managed the clinic's abortion services in both 2013 and 2015. In 2015, I also

conducted a formal interview with a doctor in the clinic and informal interviews with another doctor, a nurse, and a psychologist. It was also in the municipality where this primary care clinic was located that I was able to speak to both a doctor and a government official at the local health ministry in 2013. I transcribed my recorded interviews and took detailed ethnographic notes of my other observations within this clinic. These were then coded according to several analytical categories, including the details of clinical practice with regards to abortion, the rights discourse deployed by these health professionals, their description of the broader political context in which the clinic operated, and in 2015 the changes in clinical practice since my earlier visit and the reasons for these changes, and the new diagnostic tools in use.

Though I spoke to a number of staff at the primary care clinic that is the focus of my research, my formal interviews are the sources of the majority of the direct quotes I use in this article. My most important informant, who I interviewed in 2013 and 2015, was the social worker who ran the abortion-counseling service at the clinic, and who seemed also to be influential in shaping the services offered at other clinics in the municipality. She was in her late 30s. In 2015 I was also able to conduct a formal interview with a young doctor, also in his late 30s, who had been hired at the clinic in order to provide aspiration abortion to women who were just past the first trimester. Both spoke quite openly about the legal pregnancy terminations they were providing and expressed no concerns about any possible legal repercussions. The insider activists treated patients seeking abortion in a busy primary care clinic alongside patients seeking many other kinds of care. These abortions were extensively documented as legal pregnancy terminations, with paperwork filling large folders stored as any other patient records would be.

In 2013 this primary care clinic was one of seven clinics listed on municipal health ministry documents as providers of abortion-counseling services. By 2015 the social worker I spoke to said that this clinic and one other had shifted from abortion counseling to legal pregnancy termination, a service they were able to offer because both clinics had physicians on the health care teams that handled the abortion service, who were able to prescribe misoprostol. The social worker said "at least" three other clinics continued to provide abortion counseling, because their health care teams did not include medical doctors so they were limited to advising women on the appropriate use of misoprostol. She said some other clinics that had offered counseling in 2013 had discontinued the practice. At the time of my interview with her in 2015, the team that did legal pregnancy termination in her clinic consisted of two social workers, two psychologists, and one doctor.

This paper analyzes the changing strategies of activist health professionals working within the government health system to expand the availability of abortion. The sociology of social movements has often focused on a false dichotomy between activists and the "insiders" within state structures whom they seek to influence, ignoring the key role played by "insider activists" who work within the state apparatus [30]. It is this "insider activism" that I focus on here. This activism has special implications, because medical professionals who are government employees are using the state health apparatus to help expand women's access to safe abortion.

## Results

Seeking an alternative that would overcome the limitations of abortion counseling, the health professionals in this clinic developed and codified the legal pregnancy termination approach. Abortion counseling opened the door to legal pregnancy termination in two key ways. First, the revolutionary act of listening to women's narratives of unwanted pregnancy forced these insider activists to rethink the implications of abortion access in patients' lives. Second, abortion counseling required a procedure to determine which of the presenting cases could be defined as legal pregnancy terminations based on the exceptions written into the law.

Already in 2013, the insider activists I spoke with were finding that conducting abortion counseling forced them to institutionalize an approach to abortion on legal grounds that was nonexistent in clinic regulations, vaguely defined in health ministry guidelines, and had been completely missing from their professional training. These insider activists found that both the rape exception and the health exception were open to a wide range of interpretations, and their understanding of these laws changed as they listened to women's stories and considered the structural constraints that each woman faced in her daily life. As the social worker said in a 2013 interview, "It was a learning experience for us because it never happened that a woman would arrive [at the clinic] and say, 'Hi, how are you, I'm here to request a legal pregnancy termination.'" The grounds for a legal exception were not among the questions asked initially as part of the abortion counseling, but details that suggested these legal grounds would come out in the narrative of the undesired pregnancy. In 2013 the social worker described an early case that started clinic staff thinking about the need to specifically investigate this element of the pregnancy narrative:

Then another girl who came to the counseling, who, like all our patients was pregnant and didn't want the baby and in her story she starts to tell us, when we talk about how many weeks gestation, diagnosing the

gestational age, she says, "Oh, but I know exactly, exactly, exactly when I got pregnant." And, it was because she had been raped while out in the street. And with this girl's story we were like, "Ah!" You could say the light bulb went on over our heads, "So *this* is a legal pregnancy termination, right!"

In the process of counseling women and girls seeking to terminate pregnancies, the health professionals were forced to realize that neither they nor their patients had a clear idea of what legal grounds for pregnancy termination looked like in practice.

Forced to consider this question for the first time, the health professionals began to institutionalize the exceptions that make an abortion legal by using women's stories as their point of departure. Already in 2013 this had begun to result in broader interpretations of these legal grounds, as clinic staff discussed each case. The social worker explained in the same 2013 interview that they did not define rape solely based on the presence of physical force. They were defining rape as involving many kinds of coercion or pressure, including "a woman who is forced by her ex-partner, for example, in order to get money to feed her kids, to exchange sexual favors for child support." Their work was aided by the fact that the 2012 Supreme Court decision clarifying the rape exception had explicitly said that in order for rape to be claimed for a legal pregnancy termination, the rape charges did not have to be filed with police. The ruling specifically stated that a sworn statement from the victim claiming rape was sufficient [2].

### Creating a diagnostic instrument for legal pregnancy termination

By the time I returned to this clinic in 2015, what had begun as a halting process of slowly extending the definition of abortion on legal grounds to include cases that required a broad interpretation of the criteria, had become a fully institutionalized new diagnostic process, with a diagnostic instrument created by clinic staff. This instrument was an intake form that was completed with the patient and was used to interpret essentially all requests for abortion as legitimate based on legal grounds. A psychologist who worked in the abortion service at this clinic described in 2015 how at first they thought of abortion as a crime, so their approach was a harm reduction strategy, but later they came to think of abortion "not as a crime, but as a health concern." When I spoke to the social worker in 2015, she described how at the time I interviewed her in 2013 they had only handled a few abortions as legal pregnancy terminations. In 2014, out of 180 patients who went through abortion counseling in the clinic, 140 were handled as legal pregnancy terminations with clinic staff directly providing

misoprostol. And so far in 2015, "practically out of 5 women that we're able to see each week, all 5 are legal [pregnancy] terminations." For a quantitative analysis of the expansion of this kind of legal pregnancy termination practice at similar clinics in Buenos Aires, see the recent piece in the *International Journal of Gynecology and Obstetrics* [31].

Indeed, the fact that elements of women's stories were crucial to decide how to "diagnose" the legality of the pregnancy termination is not surprising. This constitutes a form of medical history taking, though in this particular case the diagnosis involves a legal judgment and not merely a medical one. There are debates about the best ways to take medical histories so that patients are able to fully share their perspectives [32]. But even those who suggest that patients must be allowed to play an active role in constructing their medical history have not suggested that the history would fundamentally change the diagnostic categories that the health professional must consider. Fitting the narrative to the appropriate diagnostic categories is the task of a trained medical professional with the expertise to make those judgments. In this clinic it was patients' experiences that were then translated into diagnostic criteria, their narratives turned into a powerful instrument for expanding abortion access.

Between 2013 and 2015 the insider activists created a diagnostic instrument by incorporating the stories women told about how and why pregnancy disrupted their lives, livelihoods, and well-being as categories that clearly suggested that negative health effects were likely. Rather than fitting the story to pre-existing diagnostic categories, they constructed the diagnostic categories around women's stories. The social worker described this process, while holding up a copy of the diagnostic tool, as follows:

These indications [of legal pregnancy termination], women taught them to us. Which are the indications that women come to us with… We didn't make this up. It was a conversation… 'Hey, women say this or that thing a lot!' [For example,] they don't have a partner with whom to raise the child. So that must be important. They often say that they're still trying to finish high school. Or they're in Women Do It [*Ellas Hacen*, a workfare program]. So we thought about what things they say often. Or we began to see, when we systematically analyzed the medical histories, that many who wish to terminate a pregnancy are still in the postnatal period [from a previous birth]… The act of listening to women, together with our theoretical frame, gave rise to this list of indications.

So, in contrast to the typical taking of a medical history and making a medical judgment about what

diagnostic categories are applicable, this was a process in which a diagnostic instrument was created from categories that arose from patients' narratives.

In 2015, this approach of legal pregnancy termination had culminated in the creation of intake forms, which were being utilized in what was described as a pilot phase of the new diagnostic instrument. These forms included a checklist of possible categories that might result in designating the case as a legal pregnancy termination (see Additional files 1 and 2 for English translations of these intake forms). This checklist included the headings: direct risks to health, indirect risks to health, mental health risks, and socio-familial determinants that together with the pregnancy pose a risk to the woman's health.

By creating these forms, the clinic staff had taken women's stories and created a diagnostic instrument that legitimized their reasons for needing to terminate a pregnancy. The health care professionals described the creation of these forms as part of a process of creating objective criteria for deciding these cases, after considering ways to remove subjective judgments about which cases fit the criteria:

> What criteria are we using? Sure, if she cries and she seems distressed she must be very affected [by the situation]. But, well, what happens with the woman who doesn't cry in the interview, who is distressed but doesn't show it? So the challenge these last few months, and which is now in a pilot phase, of the diagnostic instrument, where we can make sure these criteria are objective, where we make a diagnosis of the three health areas that are affected by the pregnancy.

Once the health professionals in these clinics deemed a case to be a legal pregnancy termination, they then provided misoprostol directly to women who were still in the first trimester, so that they could induce their abortion at home. In 2015 the clinic staff had begun scheduling women who were just past the first trimester for aspiration abortions with a trained physician who visited the clinic biweekly.

The health care teams also produced case histories detailing each patient's situation and the reasons for the procedure's legality under the health or rape exceptions. The social worker read to me from one of the completed legal pregnancy termination narratives:

> Patient X finds herself in a socio-familial-emotional situation whose conditions make it impossible for her to continue with this pregnancy and face the prospect of having another child. This means she is confronted with a dilemma that can only be resolved by terminating the gestation. For these reasons, which have been

probed with the diagnostic instrument, it is judged that this pregnancy is a risk for the complete health [*salud integral*] of Patient X. And this risk cannot be avoided by any means other than the legal termination of the pregnancy.

The official paperwork prepared for each patient was kept on file at the health center.

Ideas of autonomy and empowerment were central to the ways these insider activists understood of the experience of women seeking abortion in this clinic. In 2013 the social worker described the work of abortion counseling as opening opportunities for women to exercise autonomy and become empowered. She described the situation where women are denied abortion by saying:

> In order for a woman to exercise her autonomy, at a minimum she has to have two options. If, when facing an involuntary pregnancy [*embarazo involuntario*], the only option she has is to continue and live a forced maternity or, the other option, is to terminate [the pregnancy] but at the cost of risking her health or her life, or suffering psychologically or physically, that's not an option.

In contrast to women disempowered by the denial of abortion, women who were helped by the clinic:

> …come out empowered [*fortalecidas*], which is to say that this experience can also be something that provokes a kind of personal growth, increased autonomy, a way to pull away from a relationship that subjugated them, to be able to move out of their parents' house, just another decision. We really see that many women come out of this with a kind of learning experience.

If a poor woman is unable to secure a safe abortion, she may instead face what activists in Argentina call "forced maternity." This forced maternity may be accompanied by a series of concomitant restrictions in future life choices. As detailed in the diagnostic instrument, depending on the particular details of a woman's life, these can include an inability to complete schooling, family rejection or homelessness, continued contact with an abusive partner who is the parent of the child, or job loss if maternity leave protections are inadequate. These insider activists seem to have intuited through their interactions with women seeking abortion the same patterns that the Turnaway Study is now demonstrating empirically [16–19].

It may seem odd that government employees in a country where abortion is typically understood to be illegal in most cases would openly perform abortion in a

state health clinic. But that was the case in this clinic. In fact, these insider activists operated quite openly within their municipal context. They had posted a hand-written sign in one primary care clinic waiting room that advised women: "Services for Women Facing Abortion. In any case it is *your decision* and you have a *right* to protect your health. Please come as soon as possible, walk-in basis, Wednesdays at 9:00 am" [emphasis in the original]. The health professionals running this service did not express to me any concern about prosecution. The concerns they did express included demands for greater support for their work. They were lobbying the municipal health ministry to procure mifepristone[5] so that they could provide pharmaceutical abortion of equal quality to that provided in countries where the procedure is generally legal, even though mifepristone is not commercially available in Argentina. Furthermore, they were lobbying for funds to hire one more physician, which would allow them to expand their provision of legal pregnancy terminations.

The insider activists I spoke with often seemed to alternate between asserting that the legal pregnancy terminations they performed were clearly legal under the current law and also required by medical ethics that recognize women's autonomy, on the one hand, and a recognition that their view was a minority perspective and that they were engaging in insider activism meant to change the paradigm of abortion provision, on the other. The physician I interviewed described his own training as an obstetrician/gynecologist, where a medico-legal professor, "said that abortion was illegal, that it couldn't be done, unless the woman is dying, and can't be saved by any other means. That's the class I got on legal medicine. That was the interpretation." Yet earlier in the same interview he had asserted that abortion in Argentina, "is not legislated, the law doesn't say what we can't do, so we can do all kinds of things." At another point he seemed to recognize his role as an insider activist when he described his work providing abortions in the government health system by saying, "From within the system we have to find the cracks and break things open."

## Discussion
The bulk of the data I present here on clinical practice of legal pregnancy termination using this diagnostic instrument comes from one single clinic, yet there were signs that the idea of codifying legal pregnancy termination using this method was catching on. In 2015 there was one activist-run free clinic and another activist-run private clinic, which charged women on a sliding scale, both of which had begun to provide legal pregnancy terminations subsequent to the practice being adopted at the clinic I describe here. At both of these clinics, they were using intake forms that were extremely similar to

the diagnostic instrument in the government clinic. Despite the fact that similar diagnostic forms were in use in other places, the question of legality seemed to be more fraught in other contexts. The private clinic, which had only recently begun to see patients, was in a building where visitors were buzzed in after identifying themselves over an intercom, and the health professionals who worked there described setting intake and procedure appointments on separate days for each patient as a precaution against being raided by someone posing as a patient.

The limitations of these data include the fact that I did not interview women using these abortion services directly. It is possible that what appears from a provider's perspective as empowering may be experienced very differently by the women who attend the service. But it is important to consider the broader context in which women are accessing abortion in public health clinics, where only a few years earlier they would have been turned away and left with only dangerous back-alley alternatives. This research is also based on a small number of interviews, based primarily around a single primary care clinic. There is no doubt that a broader study that included more clinics and more providers would be a worthy undertaking. This research is difficult to do in a context where the legality of this practice is very much up for debate. But the abortion practices, and especially the changes in practice and the rationale for these changes, are worth considering in their own right. My own position as an academic arriving from the USA may have affected the quality of the information that interviewees were willing to share with me. But it is important to note that the interview data are corroborated by analysis of the documents used in the clinic.

## Conclusions
In the confines of a few clinics, these insider activists are creating a right to abortion in a country where most health professionals fail to recognize any such right. These insider activists further empower women by adapting clinical practice to fit their lives and choices, the opposite of the typical process of medical and legal authorities in a country where abortion is legally restricted and stigmatized.

How does this case inform our understanding of how empowerment relates to women's reproductive health? The case of unsafe abortion in countries where the practice is illegal in itself is an important reminder that women's empowerment can vary greatly depending on factors like socioeconomic status and access to reproductive health care. This suggests a need to disaggregate national measures of women's empowerment to be sensitive to the particular role that reproductive health care and reproductive decisions play in women's lives, and to employ the lens of intersectionality to understand how

factors like socioeconomic status play a role even in countries where women seem to be empowered overall. Though the reach of the clinical practice described here is small, the fact that health providers are asserting women's right to access safe abortion as fundamental for their ability to exercise autonomy and feel empowered is important.

What lessons might health professionals and policymakers interested in women's health and empowerment take from this case? First, it is important to understand that the legal status of a medical procedure and access to that medical procedure are two distinct issues. The doctor I interviewed in this clinic was quick to point out that a law legalizing abortion in the first trimester could actually restrict their work if it more clearly specified gestational limits or imposed other burdens to access, since the current, relatively vague law allowed doctors a great deal of latitude. He also pointed out that a broad legalization would mean very little if there were no doctors trained in or willing to perform abortions. On the other hand, even in legally restrictive contexts, when health professionals and activists work together their combined efforts can be very powerful, especially if they shape their clinical practice around recognizing women's autonomy and empowering them to choose the reproductive outcomes that work for them.

Questions remain about the possibilities and limits of this kind of abortion activism. When I was in Argentina in 2015, there were reasons to believe that this movement might have far-reaching impacts. A glimmer of hope is visible if we look across the Río de la Plata delta to Uruguay, where abortion counseling gave way to abortion legalization in 2012 [33]. On the other hand, at the end of 2015, national elections brought right-wing candidates to power in Argentina, including to the offices of the presidency, the governorship of Buenos Aires, and the mayor of the municipality where this clinic is located. The new president, Mauricio Macri, had been outspoken in his opposition to abortion in his former role as mayor of the city of Buenos Aires. Sometimes changes in political power have a muted effect on Argentina's large public health system, where doctors often practice, for better or worse, with a great deal of autonomy. But cuts to social services and layoffs of state employees were among the new government's key policies [34]. Yet so far, as of September of 2016, the insider activists and broader abortion rights movement have shown no signs of backing down in conflicts with conservative authorities. Whatever their fate in Argentina, the model used by these insider activists points to important strategies for empowering women through reproductive health services that are open to their stories and prioritize their health and lives in ways that are sensitive to social context.

## Open peer review

Peer review reports for this article are available in Additional file 3.

## Endnotes

[1]In the original: *consejería de pre- y post-aborto.*

[2]For more information on the abortion-counseling model, and its origins in Uruguay where it preceded the legalization of abortion in 2012, see the excellent pieces in a special issue of the *International Journal of Gynecology and Obstetrics* on Reducing Maternal Mortality by Preventing Unsafe Abortion: The Uruguayan Experience [31, 35–42].

[3]The standard procedure for pharmaceutical or medication abortion is a combination of mifepristone, which loosens the uterine lining, followed 1–2 days later by misoprostol, which causes uterine contractions. This full regimen is most effective. But in places where abortion is legally restricted, misoprostol is regularly taken by itself, though it is less effective.

[4]In the original: "no exige la configuración de un daño, sino su possible ocurrencia."

[5]Mifepristone is the other half of the pharmaceutical abortion regime, typically used in combination with misoprostol.

## Additional files

**Additional file 1:** Translation of 2-page clinic intake form. (PDF 34 kb)

**Additional file 2:** Translation of 3-page clinic intake form. (PDF 97 kb)

**Additional file 3:** Open peer review. (PDF 159 kb)

**Funding**
This article is part of a special issue on women's health and empowerment, led and sponsored by the University of California Global Health Institute, Center of Expertise on Women's Health, Gender, and Empowerment.

**Availability of data and materials**
Not applicable.

**About this supplement**
This article has been published as part of BMC Pregnancy and Childbirth Volume 17 Supplement 2, 2017: Special issue on women's health, gender and empowerment. The full contents of the supplement are available online at https://bmcpregnancychildbirth.biomedcentral.com/articles/supplements/volume-17-supplement-2.

**Authors' contributions**
The author conceived the study, collected and analyzed the data, and wrote the manuscript. The author has read and approved the final manuscript.

**Ethics approval and consent to participate**
This research was conducted with approval of the Institutional Review Boards (IRBs) of the University of Wisconsin — Madison, Luther College, and the University of Wisconsin — La Crosse. With approval of the IRBs, oral informed consent was obtained from participants.

**Consent for publication**
Not applicable.

**Competing interests**
The author declares that she has no competing interests.

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
