## [Open peer review. (PDF 159 kb) · BMC Pregnancy and Childbirth]

Reviewer reports

Title: Abortion as empowerment: reproductive rights activism in a legally restricted context

Reviewer 1: Teresa DePineres

- Major Compulsory Revisions

- You state that the law says that abortion is illegal when it does not say this explicitly, I would argue that abortion is indeed legal, but the law is interpreted narrowly, thus the providers you speak of are actually interpreting the law broadly, which is most beneficial to the woman's health, and by doing so, not breaking any law. As you state later, (Lines 455-456) "...in a country where abortion is typically understood to be illegal in most cases."
 - Related in Lines 471-472: "As in all countries where abortion is illegal," really should at least include something like, "or interpreted as illegal."
- The question posed by the author is not well-defined
- The analysis of the data is not well-defined
 - I am not familiar with Brown, et al.'s social diagnosis model, described in the manuscript, but despite the explanation of the model, there is no description regarding the structure or process of how the actual analysis was performed
- I find it hard to separate the 3rd factor of 'external actors' and the 1st factor of 'social forces.' While they are very concretely different concepts, the fact that these documents were created leads to completely different social forces, which allow providers to work
- I would argue that the factors that you outline in the final section of your "results" regarding actors beyond health professionals, the actors who you refer to are actually health professionals who just happen to be in the ministry of health
- Not sure how you define "institutional activists" or 'activist professionals," and how they are different.
- The conclusions are not adequately supported by the results
 - I understand that the "social forces" that you are trying to outline are those of feminist theory, however, I do not see evidence of how you draw this conclusion.
 - For example, Lines 457-459: "We can understand these clinics as a place where abortion's presumed illegality converges with a feminist movement that includes health professionals in its ranks and has prioritized expanding access to abortion." This statement appears that the clinic is run by feminists who are allowing physicians to participate, when you infer previously that physicians are working within the public center (not understood to be feminist-run) or privately. If this is your understanding, it would help to understand how you came to this conclusion. It is a bit confusing when you also speak to the public health concerns of the physicians-a concept that is not based in feminist theory, per se.
 - In this section, you also imply that "stories" are the reason that this is more

of a feminist model and (maybe) not a medical model, however stories are the principal component of a medical history

- In your conclusions (Lines 563-564), you state: “These activist professionals are openly making two key arguments about abortion. Their central argument is that the ability to choose abortion is a public health issue, and consequently restricting this right necessarily jeopardizes women’s health.” and “ “In order to make this argument they implicitly frame the abortion decision as a key moment in women’s lives. However, you do not provide sufficient evidence (which would include direct statements of the case made by your interviewees) to draw this conclusion.
 - You also speak of empowerment, but give no evidence from interviews of this as a concept.
- The limitations of the work are not stated
 - Other comments:
 - Lines 273-328 are more background, and not results of your analysis
 - Regarding the WHO definition of health, defined by the WHO, I think that this is more part of your background than of your results

- Minor Essential Revisions

- The term, “life project” is more of a direct translation from Spanish. “Life plans” is a closer English equivalent.
- Line 191-194: If abortion is legal and readily (ADD COMMA) available women are empowered to choose or reject maternity when an unplanned pregnancy is experienced. But when the procedure is illegal or unavailable (ADD COMMA) the options are more limited and more fraught.
- I think you meant International Planned *Parenthood* Federation In lines 532-533.

- Discretionary Revisions

These are recommendations for improvement which the author can choose to ignore. For example clarifications, data that would be useful but not essential.

Please note that both the comments entered here and answers to the questions below constitute the report, bearing your name, that will be forwarded to the authors and published on the site if the article is accepted.

Level of interest - An article of importance in its field

Quality of written English - Acceptable

Declaration of competing interests - I have no competing interests

Reviewer 2: Sarah Baum

This paper contributes a new perspective to the use of “health exception” for legal abortion provision in legally-restrictive settings across the world. The empowerment framework and motivation articulated by the activist health professionals in Argentina focuses women at the center of the conversation.

1. Throughout the paper, there are various words used interchangeably to describe the sample: “activists”, “health professionals”, “activist health professionals”, “health care teams” and “institutional activists”. While the blurry lines between these identities are at the essence of this paper, it would benefit the reader to explicitly say and define these actors from the beginning.

Background

2. Overall the background presents rich information, however it lacks a strong structure and organization
 - a. The author seems to be interpreting and evaluating the political environment in Argentina in this section. If these conjectures are from research conducted as part of this study, would it be possible to move some of this to the discussion?
3. The use of “freely” at the end of the first paragraph is trying to describe a service that is out in the open, but is confusing because it could refer to cost, is there a different word?
4. The research question states that the research intends to explore “the impact of the activist health care professionals on women’s ability to make reproductive choices.” However in the first paragraph of the methods, the author states that the focus of the research is “activism, professional authority and political discourse”. One seems to focus on women while the other on activists and providers. Based on the data collected, arguments presented, and Brown’s framework, the latter description in the methods seems to more accurately represent the research question. Please clarify.
5. The primary conclusion is stated in the background section, this would be better placed in the discussion.
6. Some citations are noted in the text using brackets (ie [1]) and others are superscript and the numbers do not seem to follow the same order. What do the superscripts refer to?
7. There is an existing body of literature regarding expansion of abortion access under the health exception
 - a. <http://www.ncbi.nlm.nih.gov/pubmed/23245405>
8. The author asserts that provision of care shifted in Argentina from 2013 to 2015. These dates seem to be identified based on the authors visits to the country, but this is not explicit. If the paper intends to compare these two dates, it would help to further describe this comparison in the methods.

Methods

9. The “activist” and “health professionals” are kept separate in the list of interviewees in the first paragraph, and the second paragraph states that the author will focus on the “insider activism” of activist health professionals. Please clarify.
 - a. Were all activists who participated working at an activist-run clinic?

- b. What types of health professionals? (ie: docs, nurses, midwives, social workers)
10. Paragraph 1: How many primary care clinics/hospitals? How many activist-run clinics?
 11. Paragraph 2: Are the Mendoza and Cordoba interviews included in analysis?
 12. Paragraph 3: When discussing the use of Brown's model of social diagnosis, you describe the "creation of a diagnostic instrument for declaring abortion legal in a context of illegality". While abortion was not provided under the exceptions for many years, would you describe the current context as one of "illegality"? Wouldn't it be more accurate to describe as highly restricted?
 13. The first part of the results describes the transition from pre/post ab counseling to a legal pregnancy termination framework. Is this completed by comparing interviews from 2013 to 2015 or by asking participants in 2015 to reflect on the transition or both? Please clarify the use of interviews over time.
 14. The methods section does not describe the analysis process: How was analysis conducted? Were the interviews transcribed? Translated? Coded? Was a qualitative software used? How did the author address any bias she brings to the analysis process?

Results

15. Overall, the results could benefit from more quotes from a variety of interview participants and even a few characteristics of the participants to know that they are distinct voices. Excerpts from the interview with the same social worker appear to be included throughout this section. It is hard to confirm whether it is in fact one social worker's interview? It is also hard to get a sense of whether she is representative of other participants' views. Would it be possible to include other voices here? If not, is her experience unique?
16. Paragraph 1: The analysis from Brown's model is not included when the author describes what will be included in the Results section. It would be useful to prepare the reader for this section here since it is a large component of the results.
17. The first two paragraphs and the majority of the third paragraph in the "From abortion counseling to legal pregnancy termination" section provide information to the reader, for example the description of the harm-reduction model, and these do not seem to be results. Is there a reason this information was not included in the Background section?
18. Were there differences in the activist run-clinics and government clinics in implementation of the diagnostic tool?
19. The Results and Discussion sections refer to "obligatory maternity", "forced maternity" and "imposed maternity", likely in various translations of the same phrase in Spanish. It may benefit the reader to be consistent.
20. In the last paragraph, it is mentioned that the provider at a particular government service did not have any fear of prosecution. Did any of the interview participants have feelings/fears about potential legal consequences at any point in this transition to legal abortion framework? How did they cope with those feelings? This would be valuable contribution for providers in other similar contexts who may have that fear.

Discussion

21. Citation for Turnaway Study is missing

22. There is a new quote introduced in the Discussion section, all new data should be presented in the Results.
23. There is existing literature discussing the harm-reduction model in other Latin American countries and broad definition of health exception such as the WHO definition briefly mentioned by the author in the results section. Both should be included in the background or discussion in order to help the reader connect the Argentine context with other locations that are using similar strategies to get women access to safe abortion despite legal restrictions.
24. Please include implications or lessons to share with other restrictive contexts.

Level of interest - An article whose findings are important to those with closely related research interests

Quality of written English - Acceptable

I declare that I have no competing interest

Response to reviewers

Julia McReynolds-Pérez
Department of Sociology
University of Wisconsin- La Crosse
La Crosse, WI 54601

Ushma Upadhyay, PhD, MPH
BMC Pregnancy & Childbirth
Special issue on women's health and empowerment

September 20, 2016

Dear Dr. Upadhyay,

I want to thank Teresa DePineres and Sarah Baum for their detailed, constructive, and insightful comments. I genuinely appreciate the feedback, and the chance to improve this article. I am attaching my revised article for publication. Based on the detailed reviewer comments, I made substantial changes to the article. I hope you will agree that this draft is a significant improvement over the earlier version.

There were three main types of changes I made that followed from the recommendations of the reviewers: Including additional information, changes in the research question and focus of the article, and changes in the organization. I will describe these three main kinds of broad changes first, and then describe some of the smaller changes that were made in response to the reviewer comments.

Additional information was added to the article as per the reviewer recommendations. The methods section now includes substantially more detail about how many interviews were conducted, in what health care settings, and with what kinds of health professionals and activists. It makes clear that the bulk of the data presented in this analysis was obtained at one primary care clinic, although similar practices were in place at other government and activists clinics, some of which I observed and others were described to me. I explained the process of interview transcription and analysis in more detail as well. I also included more direct quotes in the findings and made clearer the attribution of these quotes to a small number of professionals working in this clinic. I added details from interviews that strengthened my argument about how practice changed in this clinic, and the ways in which the practitioners were linking abortion provision to issues of autonomy and empowerment and had created their diagnostic instrument using women's stories.

I made changes to the research question to clarify that the focus of the analysis was on professional practice and not women's outcomes, since I do not have data on the latter. With the clarification of the research question I was able to focus my findings section so that it more clearly supported the central thesis of the article. One of the reviewers raised a question about the ways in which the Social Diagnosis Model was being deployed, where the distinction between the analytic categories was not clear. This question led me to revisit the Social Diagnosis Model. I decided that the distinction was actually not clear, but also that the model contributed very little to my analysis. I dropped the model from the analysis and instead focused on the substance of

the argument about changing practice in the clinic. I believe this has resulted in an article that hews more closely to the special issue's theme of women's health and empowerment, and (I hope) will be more compelling to its intended interdisciplinary audience.

I followed most of the recommendations for the reorganization of various sections of the article. I moved some material from the findings to the background, as recommended. I removed a direct quote from the discussion to the findings section. I included additional consideration of both limitations and implications or lessons in the discussion, as suggested.

Some additional smaller changes:

- I attempted to be more consistent in my use of terms regarding:
 - Force maternity
 - Insider activists
- One reviewer wondered why the text included both bracketed and superscript numbers. These are for citations and endnotes, respectively, as per *BMC Pregnancy and Childbirth* publication guidelines. I made no changes to this except that additional citations and endnotes were added.
- I dropped the use of the word "freely" to instead clarify whether I meant openly or free of charge, or both.
- I attempted to be consistent in describing abortion as either "legally restricted" or "generally understood to be illegal" rather than describing the practice as illegal, since that is precisely what is up for debate.
- I added a note clarifying why I believe that this clinical practice where women's stories are then used to craft the diagnostic tool is distinct from the typical process of medical history taking though both center around narrative accounts of patient experiences.
- I added citations regarding the harm reduction model, the origin of the practice in Uruguay, and its promotion across the region for expanding access to abortion.

Thank you, Ushma Upadhyay, for your comments and for the extension of the deadline to complete these revisions. I look forward to hearing from you.

Sincerely,

Julia McReynolds-Pérez

Reviewer reports – 2nd round

Reviewer 1: Sarah Baum

I have had the opportunity to review the revised version of the manuscript and the notes from the author. Thank you for the opportunity.

I believe the revisions greatly improve the manuscript. I have only a few minor comments for the editors to consider:

- The author asserts that abortion is safer than labor and delivery on line 247, I recommend a citation here.
- The story told on lines 372-378 seems to be from the author's own research, but this is not defined explicitly nor is the context provided such as the year. This would be helpful.
- Lines 145-153 is a description of the diagnostic tool that is described in detail in the results (sub-section starts on Line 513). Is there a reason to put this information in the background?
- Line 514: this "clinic" (grammatical edit)
- In the revised Results, there is only one sub-heading "Creating a Diagnostic Instrument for Legal Pregnancy Termination". Does the author want to include additional sub-sections or remove this heading?
- The author is introducing new data into the discussion section (paragraph starting on line 715 and line 735). These comparisons to other clinics might be better placed in the results while the discussion could focus more on interpreting the findings (such as is included in the conclusion) – but ignore if the results/discussion is used differently due to disciplinary styles